# Research on Continuous Machining Strategy for Five-Axis Machine Tool: Five-Axis Linkage to Four-Axis Linkage

**Yesong Wang [1,\*], Liang Ji [2,3], Jiashang Dong [3,4], Manxian Liu [3,\*] and Jiang Liu [3]**

1    School of Mechanical Engineering, Jiangsu University of Science and Technology, Zhenjiang 212003, China
2    Beijing Shiny Technology Co., Ltd., Beijing 100039, China
3    School of Mechanical Engineering, University of Science and Technology Beijing, Beijing 100083, China
4    Avic Jonhon Optronic Technology Co., Ltd., Luoyang 471003, China
\*    Correspondence: wangys22@just.edu.cn (Y.W.); d202210291@xs.ustb.edu.cn (M.L.)

**Abstract:** The article presents a novel strategy for enhancing the efficiency of machines that are used for complex structure machining. It proposes a low-cost five-axis four-linkage milling system as an alternative to the more expensive five-axis five-linkage system. Kinematic analysis of the machine tool is conducted to establish a correlation between the tool location data and the displacement of kinematic axes. An interpolation algorithm is then devised to determine a four-axis linkage milling strategy. The theoretical errors of the interpolation trajectory are observed to be reduced following the transformation. The research employs impeller processing as a case study, wherein the five-axis linkage machining path is translated into a more efficient five-axis four-linkage path using the interpolation algorithm. The practical application of this novel milling strategy confirms its effectiveness in processing the integral impeller within acceptable efficiency parameters. The results provide a theoretical foundation for the practical application of the low-cost five-axis four-linkage machining strategy in high-precision five-axis five-linkage machine tools.

**Keywords:** five-axis machine tool; continuous machining strategy; tool path; interpolation algorithm; cost-effective

## 1. Introduction

Compared to three-axis machining, five-axis machining provides numerous benefits, such as faster material removal rates and enhanced surface finish [1]. With the added flexibility of adjusting tool orientation through additional degrees of freedom, five-axis machining can achieve highly efficient machining. For intricate surface components like engine blades and turbine blades, the conventional three-axis surface processing technology often falls short in meeting the specific processing requirements due to their uneven spatial curvature [2]. This is where the significance of five-axis processing comes into play. Therefore, the five-axis CNC machine tools have become essential equipment in modern manufacturing, specifically for the production of intricate curved components and precision molds [3]. To fulfill the processing needs of high-precision curved surfaces, in terms of surface processing and forming requirements, a five-axis or even six-axis CNC machine tool is theoretically necessary. However, as the demand for five-axis machining tools continues to grow, the five-axis linkage control system, being the most critical component of such systems, has experienced a serious shortage.

Currently, most of established five-axis linkage CNC systems available commercially are prohibitively expensive, beyond the financial capacity of even some prominent companies [4]. The main factors behind the exorbitant pricing of established commercial five-axis linkage systems that are highly precise and stable are attributed to the steep research and development costs involved, along with the technical complexities associated with their development. Multi-axis linkage CNC programming is complex, and device development is challenging, specifically when it comes to multi-axis linkage numerical control. Currently,

multi-axis linkage numerical control systems continue to hold the sway in mainstream surface processing, which poses a predicament regarding their exorbitant pricing [5].

To reduce costs and attain comparable machining accuracy to conventional multi-axis simultaneous machining systems, there is growing interest in cost-effective multi-axis CNC systems [6]. Such systems typically encompass five or more independent axes; however, their control systems only allow simultaneous four-axis machining. Also acknowledged as 4 + 1 or 4 + 2 axis machining, this pertains to the coordinated movement of four motion axes on a five-axis or six-axis CNC machine tool, while the remaining one or two motion axes intermittently move or remain fixed at a predetermined position. This paper takes the five-axis four-linkage numerical control system as an example. Compared to the five-axis linkage system, the four-axis linkage system is 30% more cost-effective, and four-axis machining technology has found broader application in factory production. At present, many scholars have studied the application of the four-axis linkage system in surface machining [7,8].

For the complex surface machining of the integral impeller, the four-axis linkage machining strategy of Power MILL and the corresponding transformation techniques are used to process the impeller [9]. A four-axis trochoidal tool path planning algorithm utilizing a ball-end milling tool is suggested to successfully process complex curves. However, the four-axis machining in the aforementioned curved surfaces is typically specific to some particular curved surface instances and entails a certain level of skill, which hinders the practical application of cost-effective curved surface parts processing. So, it is imperative to delve into the cost-effective and high-precision four-linkage processing method using five-axis machine tools. However, conventional CAM software is incapable of producing a four-axis tool path directly for a five-axis CNC system. The approach in this paper involves designing the tool path for traditional five-axis continuous machining in standard CAM software. We then use the method outlined in this paper for translation into a five-axis four-linkage continuous machining trajectory that the device can leverage, thus accomplishing the objective of five-axis five-linkage machining. This reduction in the production cost of high-end parts carries immense significance for the low-end machine tool industry as it allows for the full potential of cost-effective five-axis CNC machine tools. This method's ability to lower the production cost of high-end parts is of paramount importance for the low-end machine tool industry as it unlocks the full cost-effective potential of five-axis CNC machines. Moreover, this method provides an alternative, feasible, and low-cost solution for small and medium-sized enterprises to manufacture parts with complex surface geometries.

At present, several research institutions and manufacturing enterprises employ NC programming software like CATIA, UG, Master CAM, and CIMATRON to undertake programming and processing of impellor parts [10]. Heigel Jarred et al. [11] investigated the five-axis NC machining technology of integral impellors and leveraged the numerical control module of standard software to execute control over the impellor and attain its NC machining tool path. Generally, the research on the five-axis machining trajectory primarily centers around refining the interpolation of the five-axis trajectory while evading potential interference and collisions. In the literature, many algorithms have been developed for tool path generation and tool-gauging avoidance to achieve highly efficient tool path planning without gauges [2,12].

Shi et al. [13] established a theoretical foundation and proposed an implementation approach for attaining a multi-axis CNC system with fewer axes, along with presenting an algorithm for a multi-axis CNC system with additional axes. However, the actualization process requires the transformation of the machine tool, and the tool path planning algorithm is intricate, leading to low processing efficiency in machining complex surfaces. The two simplest approaches for tool path generation comprise using curves of constant parameters or intersection curves of the parametric surface and a series of vertical planes. However, in both the iso-parameter and iso-plane methodologies, the machining paths do not align with a constant scallop height and, as a result, lead to inadequate surface

precision [14–16]. The iso-scallop machining method was proposed to address this issue; however, the algorithms are complicated. Literature studies have also investigated errors associated with five-axis machining processes [17,18]. For example, Li et al. [19] conducted an experimental study to test the motion error of the machine axis in the five-axis machining process; the systematic errors were identified and could be effectively compensated. High-precision axial error measurement using a frequency-modulated interferometer was developed to be a suitable solution for noncontact and high-precision spindle error measurements in the machining process [20]. Jiang et al. [21] introduced a series of groundbreaking techniques, employing deep learning and reinforcement learning technologies in five-axis machining to predict contour errors and provide compensation. These methods accounted for non-linear issues such as clearance and friction involving the feed shaft in an adaptive manner. Mikhail et al. [22] explored how surface properties could be utilized to screen for tool interference with the component surface in sculptured surface machining. The approach involves translating the tool contact points into tool position points; however, the computation is time consuming. Tran et al. [23,24] proposed algorithms for rapidly detecting and correcting collision between a manually pre-defined tool and an arbitrary workpiece. The workpiece is represented as a point cloud, and the tool is modeled through implicit equations. Although prior research predominantly gravitates towards optimizing the trajectory of a five-axis system, a few studies involve the machining trajectory transformation algorithm from a greater-axis system to a fewer-axis system. Aiming at the particular circumstance of the tool position data track in a five-axis system, Zhang et al. [25] presented a local corner transition algorithm with a global motion planning strategy that considers a more rational axial acceleration limit. Xu et al. [26] proposed a novel external local interpolation method, which used a quantized polynomial feed rate scheduling strategy to generate motion profiles along mixed tool paths to smooth discrete paths and improve machine tool motion performance.

In conclusion, The five-axis five-linkage machine tool has advantages over the five-axis four-linkage machine tool in terms of higher precision, broader machining range, and higher production efficiency; however, it also requires higher investment and operating costs. In the eyes of some small and medium-sized enterprises, the five-axis four-linkage machine tool has better cost-effectiveness. This paper uses an algorithm to convert the five-axis five-linkage machining trajectory into a five-axis four-linkage machining trajectory to achieve the previously complex parts that could only be achieved by five-axis linkage. While prior studies haven not specifically focused on this algorithmic transformation, their exploration of nonlinear errors, machining precision, detection, and simulation methods can guide the validation and implementation of the proposed algorithm in practical machining. Moreover, this paper's concept of reducing linkage axes can provide inspirational significance for lowering the cost of machining with more linkage axes.

The remainder of the paper is organized as follows. Section 2 provides an analysis of the position of the tool and the displacement of the tool axis. Section 3 shows the design of the interpolation algorithm and the procedure to convert the continuous five-axis tool path into the continuous four-axis path. Section 4 provides a case study to illustrate the effectiveness of the proposed strategy. Finally, conclusions are drawn in Section 5.

## 2. Relation between Tool Position Data and Machine Tool Axis Displacement

The tool path denotes the tool's motion trajectory in CNC machining, acquired by determining the contact point and the optimal orientation between the tool and the workpiece based on the component's geometry and process requirements. In general, a tool path is created based on the tool position data, which is typically generated using CAM software. The tool position data consist of the tool's center coordinates $(x, y, z)$ and its axis vector $(i, j, k)$. The center coordinates are used to determine the path of the machining, while the axis vector determines the tool's position. Figure 1 illustrates that the tool path comprises the trajectory of the tool's center point and the trajectory of the unit-length point on the tool axis. The vector of the tool axis is represented by the blue that links the two trajectories.

To transform a five-axis five-linkage trajectory into a five-axis four-linkage trajectory, it is essential to comprehend the correlation between the tool position data and the motion of the machine tool axes.

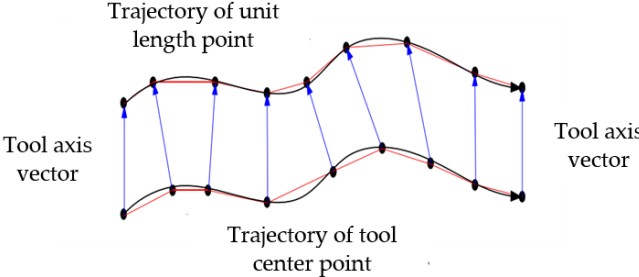

**Figure 1.** The schematic diagram of the tool path.

This paper focuses on a Sinumerik 828D controlled five-axis CNC machine tool that boasts a rotary table and a swinging head; the schematic diagram is shown in Figure 2. This machine tool is capable of simultaneously performing any four-axis linkage.

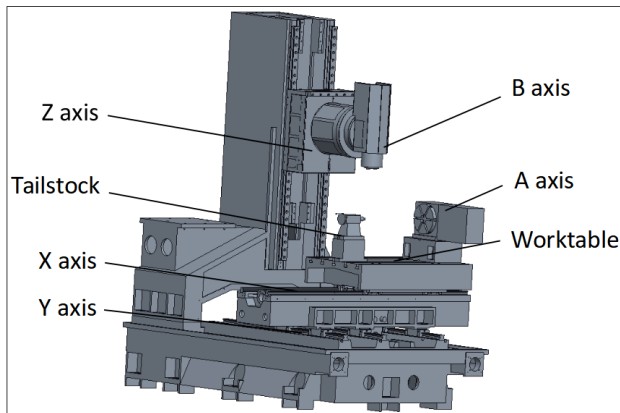

**Figure 2.** A typical five-axis CNC milling machine with a swinging head and a rotation table.

The machine tool under consideration has five axes in motion, comprising three linear axes (X/Y/Z) and two rotational axes (the tool swing shaft B and the worktable rotating shaft A). This type of machine tool is classified as a swing head and rotary table five-axis machine tool. The structure of this machine tool is intermediate, between the double swing head and the double rotary table types. Due to the workpiece rotation on the A axis, this machine tool can process workpieces of a relatively broad size range, making it a highly versatile CNC machine tool. The parameters of the machine tool are shown in Table 1.

**Table 1.** Main parameters of machine tool.

| Item | Parameter |
|---|---|
| Spindle maximum speed | 12,000 r/min |
| Spindle rated speed | 8000 r/min |
| Axis travel of X, Y, and Z | 1000/600/500 mm |
| B, A axis angle range | $\pm 90°/\pm 360°$ |
| Fast moving speed of X, Y, and Z axes | 24 m/min |
| Numerical control system | Sinumerik 828D |

*2.1. Kinematic Modeling of Machine Tool*

Machine tools with this type of structure exhibit many different types according to the order of translation of their worktable. The structure of the moving chain is shown in Figure 3. The motion of the X, Y, Z axes between the B-axis and A-axis, based on

mathematical knowledge of translation and the translation matrix, is independent of the sequence. As a result, the motion modeling of the machine tool can be classified as the following process. Then, 'A' is the A-axis (rotation on the X-axis), and 'B' is the B-axis (rotation on the Y-axis). 'A″' is the workpiece rotation axis, and 'B' is the tool rotation axis.

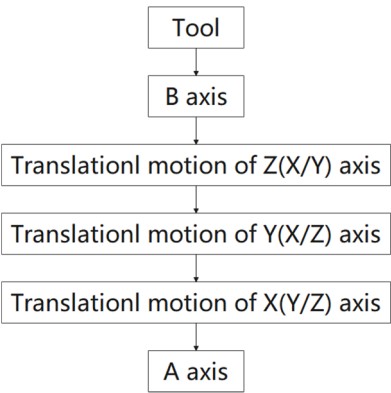

**Figure 3.** Kinematic chain of the A′-B machine tool.

In order to depict the kinematic correlation between the tool data and the NC machine tool axes, the coordinate system shown in Figure 4 is established. In this system, $O_wX_wY_wZ_w$ denotes the work coordinate system, which serves as the reference frame for the data in the location source file. $O_tX_tY_tZ_t$ represents the tool coordinate system, where the coordinate origin coincides with the location of the tool itself. Similarly, $O_bX_bY_bZ_b$ pertains to the B-axis coordinate system, where the coordinate origin is situated at the intersection of the tool axis and the B-axis. $O_aX_aY_aZ_a$ denotes the A-axis coordinate system, where the coordinate origin can be any point along the A-axis. Moreover, L represents the distance between the center of the tool and the center of rotation of the B-axis.

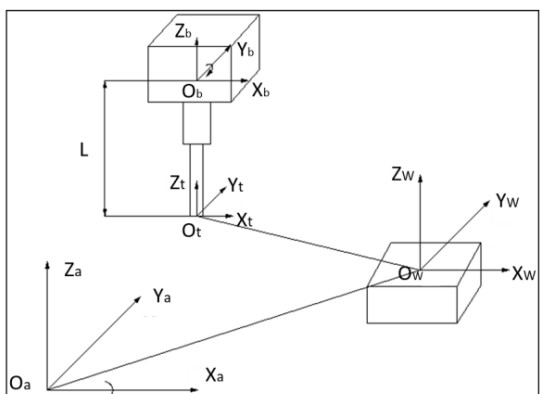

**Figure 4.** Coordinate system of the A′-B machine tool.

The process of solving the displacement of the machine tool axis is the process of decomposing the relative motion between them into each axis using a relational equation. Given that the X, Y, and Z axes translate in space between the B and A axes, and that the principles of mathematics demonstrate the sequence independence of translation and the translation matrix, The modeling of machine tool motion can be categorized into the following process.

As depicted in Figure 4, the machine tool in its initial state features a tool axis that is parallel to the Z axis. The machine tool coordinate system and the workpiece coordinate system are aligned with each other, with the coordinate origin of the tool coordinate system coinciding with that of the workpiece coordinate system. After undergoing a coordinate

transformation, the tool position data can be related to the movement of the machine tool axes in the following manner.

$$[i, j, k, 0]^T = T(r_{m1}) \cdot R_X(-\theta_A) \cdot T(r_s - r_{m1} + r_{m2}) \cdot R_Y(\theta_B) \cdot T(-r_{m2}) \cdot [0, 0, 1, 0]^T \quad (1)$$

$$[x, y, z, 0]^T = T(r_{m1}) \cdot R_X(-\theta_A) \cdot T(r_s - r_{m1} + r_{m2}) \cdot R_Y(\theta_B) \cdot T(-r_{m2}) \cdot [0, 0, 0, 1]^T \quad (2)$$

In Equations (1) and (2), $T$ and $R$ signify, respectively, the homogeneous coordinate transformation matrices for translational and rotational motion, which are commonly used in computer graphics.

$$T(r_{m1}) = \begin{bmatrix} 1 & 0 & 0 & m_x \\ 0 & 1 & 0 & m_y \\ 0 & 0 & 1 & m_z \\ 0 & 0 & 0 & 1 \end{bmatrix}$$

$$T(r_s - r_{m1} + r_{m2}) = \begin{bmatrix} 1 & 0 & 0 & X - m_x \\ 0 & 1 & 0 & Y - m_y \\ 0 & 0 & 1 & Z - m_z + L \\ 0 & 0 & 0 & 1 \end{bmatrix}$$

$$T(-r_{m2}) = \begin{bmatrix} 1 & 0 & 0 & 0 \\ 0 & 1 & 0 & 0 \\ 0 & 0 & 1 & -L \\ 0 & 0 & 0 & 1 \end{bmatrix}$$

$$R_X(-\theta_A) = \begin{bmatrix} 1 & 0 & 0 & 0 \\ 0 & \cos(\theta_A) & \sin(\theta_A) & 0 \\ 0 & -\sin(\theta_A) & \cos(\theta_A) & 0 \\ 0 & 0 & 0 & 1 \end{bmatrix}$$

$$R_Y(\theta_B) = \begin{bmatrix} \cos(\theta_B) & 0 & \sin(\theta_B) & 0 \\ 0 & 1 & 0 & 0 \\ -\sin(\theta_B) & 0 & \cos(\theta_B) & 0 \\ 0 & 0 & 0 & 1 \end{bmatrix}$$

By substituting Equations (1) and (2) into the transformation matrices $T$ and $R$, Equations (3) and (4) express the motion transformation relationship of the five-axis machine tool.

$$\begin{bmatrix} i \\ j \\ k \\ 0 \end{bmatrix} = \begin{bmatrix} \sin(\theta_B) \\ \sin(\theta_A)\cos(\theta_B) \\ \cos(\theta_A)\cos(\theta_B) \\ 0 \end{bmatrix} \quad (3)$$

$$\begin{bmatrix} x \\ y \\ z \\ 1 \end{bmatrix} = \begin{bmatrix} -\sin(\theta_B) \cdot L + s_x \\ -\sin(\theta_A) \cdot \cos(\theta_B) \cdot L + \cos(\theta_A) \cdot (s_y - m_y) + \sin(\theta_A) \cdot (s_z - m_z + L) + m_y \\ -\cos(\theta_A) \cdot \cos(\theta_B) \cdot L - \sin(\theta_A) \cdot (s_y - m_y) + \cos(\theta_A) \cdot (s_z - m_z + L) + m_y \\ 1 \end{bmatrix} \quad (4)$$

## 2.2. Displacement of the Machine Tool Axis

With the kinematic model of the machine tool established, we can use the tool position data to determine the moving component of each axis of the machine tool. It is important to note that the angle solution of the two rotary shafts is not necessarily unique, which means that a particular sequence of analysis must be followed to compute the values of each moving axis. Based on the information presented in Table 1, the feasible range of the B-axis is determined. By solving Equation (3), we can determine the moving component of the rotating axis B of the machine tool (as shown in Equation (5)). We can also calculate the angle of the B-axis using Equations (3) and (5), as expressed in Equation (6). When

addressing the angle solution of the A-axis, the analysis is broken down into the following scenarios (as presented in Equation (7)).

$$j^2 + k^2 = \cos(\theta_B)^2 \tag{5}$$

$$\theta_B = \arctan\left(\frac{i}{\sqrt{(j^2 + k^2)}}\right) \tag{6}$$

$$\theta_A = \begin{cases} \arctan(\left|\frac{j}{k}\right|) \ (k > 0, j > 0) \\ \pi - \arctan(\left|\frac{j}{k}\right|) \ (k < 0, j > 0) \\ \frac{3}{2}\pi - \arctan(\left|\frac{j}{k}\right|) \ (k < 0, j < 0) \\ 2\pi - \arctan(\left|\frac{j}{k}\right|) (k > 0, j < 0) \\ \frac{1}{2}\pi \ (k = 0, j > 0) \\ \frac{3}{2}\pi \ (k = 0, j < 0) \\ 0 \ (k > 0, j = 0) \\ \pi \ (k < 0, j = 0) \end{cases} \tag{7}$$

It is important to note that the solutions for both the B-axis and A-axis are represented in radians and need to be converted to their corresponding angle values.

$$\text{beta} = \frac{\theta_B \times 180}{\pi}$$

$$\text{alpha} = \frac{\theta_A \times 180}{\pi}$$

Once the angle of the rotary axis has been determined, the displacement of each linear axis can be obtained by Equations (8)–(10).

$$X = x + \sin(\theta_B) \times L \tag{8}$$

$$Y = m_y + \cos(\theta_A) \times (y - m_y) - \sin(\theta_A) \times (z - m_z) \tag{9}$$

$$Z = m_z - L \times (1 - (\cos(\theta_B))) + \sin(\theta_A) \times (y - m_y) + \cos(\theta_A) \times (z - m_z) \tag{10}$$

Through the process of creating and solving the kinematics model, it is possible to derive the kinematic displacement and rotation angle for each axis of the machine tool, with the help of tool data.

## 3. Four-Axis Machining Strategy of Five-Axis Machine Tool

Based on the analysis of the processing method of the five-axis machine, it is recommended to obtain the five-axis four-linkage path of the tool path using a tool path algorithm. To achieve this objective, it is crucial to consider the structure of the machine tool and the features of the parts being machined. Thus, the entire four-axis machining strategy of the five-axis machine tool should be equalized and equated, with a focus on three key steps.

Firstly, it is important to select representative components featuring complex surfaces, analyze their geometric models, and assess their machining process requirements, while also evaluating the part's processing sequence. Once the processing sequence has been determined, the five-axis tool path for machining the part needs to be generated using NC programming within the CAM software. Post successful track detection, the tool position file can be extracted.

Secondly, it is essential to evaluate the five-axis linkage tool path carefully, determining the correlation between tool data and the translating axes of the CNC machine tool, to transform the five-axis five-linkage tool path into a five-axis four-linkage tool path.

Lastly, the transformed tool path needs to be analyzed and validated by theoretical evaluation and NC machining simulation. The comprehensive strategy diagram for the four-axis linkage machining approach of the five-axis machine tool is detailed in Figure 5.

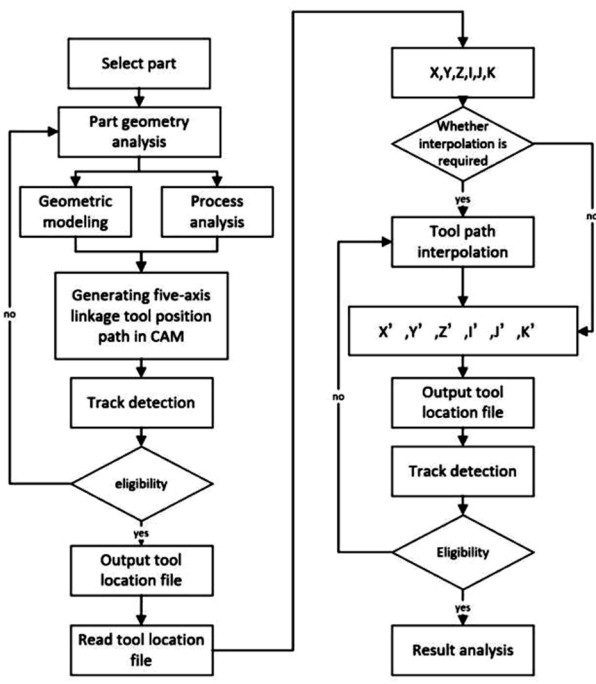

**Figure 5.** Comprehensive strategy diagram of the four-axis machining approach of the five-axis machine tool.

### 3.1. Interpolation Algorithm

The fundamental concept underlying the tool path interpolation algorithm is to identify and determine one or more interpolation points located between two adjacent points on the five-axis tool path. This method aims to transform the motion of the machine tool between trajectory points from a five-axis linkage trajectory into a four-axis linkage trajectory. This requires the motion of a certain axis of motion to be determined by the point of interpolation. As outlined in the second section, the positional value of the tool center is influenced by the rotational and linear axis motion, while the rotational axis movement solely affects the size of the tool axis. When the movements of a specific linear axis are restricted, the impact of the rotation axis on the positional coordinate cannot be controlled, resulting in significant alterations to the tool path trajectory. If the rotation of a rotating shaft is restricted, the change in the position of the center of the tool caused by the rotating shaft can be compensated for by other linear axes.

In summary, the fundamental concept behind the interpolation algorithm is to interpolate between two points on the five-axis trajectory. By constraining one of the rotational axes, the five-axis trajectory can become a four-axis trajectory. However, as the focus of the tool path algorithm is the CAM software's tool path file, the comprehensive planning of the algorithm can be divided into three core parts.

Firstly, it is necessary to analyze the format of the tool file exported by the CAM's automatic programming software and extract the tool position data from the tool file. Secondly, it is crucial to judge whether adjacent tool sites are interpolated. Subsequently, calculating the tool center coordinates and tool axis vector is essential, and the interpolated data need to be integrated to simulate the interpolated tool path. Finally, after processing the new tool data, it is essential to write it into the tool position file following the structure of the initial tool position format, resulting in the creation of a new tool position file. Figure 6 illustrates the process of implementing the interpolation algorithm.

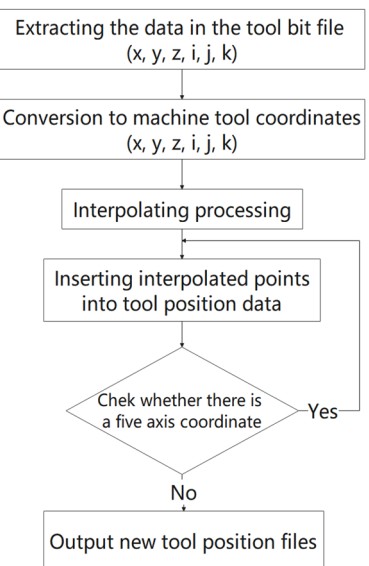

**Figure 6.** The implementation process of the interpolation algorithm.

To ensure the continuity and smoothness of tool path, it is vital to utilize interpolation theory to interpolate the linear tool center point and rotate the axis by employing the point comparison method and maintaining even step sizes. To interpolate NC machine tools, the linear interpolation method is typically utilized. In order to ensure continuity of the interpolation trajectory, linear interpolation is carried out for the center point of the tool. At the same time, to facilitate the transformation from five-axis linkage to four-axis linkage, one rotation axis remains unchanged during interpolation, while another rotational axis rotates by half the difference between adjacent five-axis points. The number of interpolation points has a direct impact on processing efficiency. In order to align the coordinates of the five-axis machine tool to the four-axis machine tool, the number of interpolation points is 2.

The schematic diagram displaying the tool center point interpolation process is available in Figure 7a. The five-axis tool positions M and N require the addition of two new points, $C_{11}$ and $C_{21}$. The tool position data for tool position M are given by $(x_0, y_0, z_0, i_0, j_0, k_0)$, while the tool position data of the tool position N is $(x_1, y_1, z_1, i_1, j_1, k_1)$. To guarantee an even step length, the interpolated points $C_{11}$ and $C_{21}$ should be uniformly distributed along the linear interpolation paths of M and N. The tool axis vector interpolation process is shown in Figure 7b.

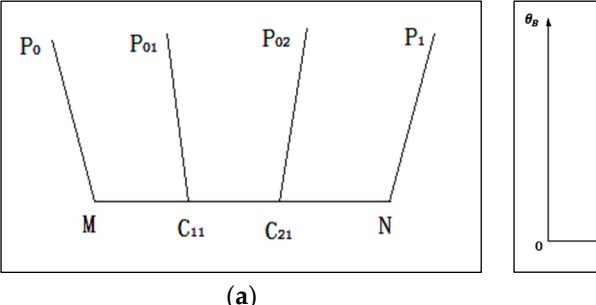

| (a) | (b) |

**Figure 7.** Schematic diagram of the interpolation. (**a**) The point interpolation of the tool center (**b**) The tool axis vector interpolation.

The axis displacement values for tool center points M and N are $(X_0, Y_0, Z_0, B_0, A_0)$ and $(X_1, Y_1, Z_1, B_1, A_1)$, respectively. The angle values for M, $C_{11}$, $C_{21}$, and N are $(B_0, A_0)$, $(B_0, 1/2(A_1 - A_0) + A_0)$, $(B_1, 1/2(A_1 - A_0) + A_0)$, and $(B_1, A_1)$, respectively.

After interpolation, the track points M and N that formerly exhibited adjacent five-axis linkage will be transformed into a five-axis four-linkage trajectory: M, $C_{11}$, $C_{21}$, and N. The coordinate values of these points are as follows.

$$N10\ X_0, Y_0, Z_0, B_0, A_0$$

$$N20\ X_{c1}, Y_{c1}, Z_{c1}, B_0, \frac{1}{2}(A_1 - A_0) + A_0$$

$$N30\ X_{c2}, Y_{c2}, Z_{c2}, B_1, \frac{1}{2}(A_1 - A_0) + A_0$$

$$N40\ X_1, Y_1, Z_1, B_1, A_1$$

### 3.2. Solution of the Interpolation Point

The process of solving the tool position data of the interpolation points $C_{11}$ and $C_{21}$ is as follows. To find the tool center point data of the interpolation point, two points are inserted between M and N to ensure the continuity of the trajectory of the tool center. The position of the tool center of the interpolating point must be on the line of the M and N point. It is assumed that the point coordinates of $C_{11}$ and $C_{21}$ are, respectively, $(x_{C1}, y_{C1}, z_{C1})$ and $(x_{C2}, y_{C2}, z_{C2})$.

$$x_{C1} = x_0 + \frac{1}{3}(x_1 - x_0) \tag{11}$$

$$x_{C2} = x_0 + \frac{2}{3}(x_1 - x_0) \tag{12}$$

$$y_{C1} = y_0 + \frac{1}{3}(y_1 - y_0) \tag{13}$$

$$y_{C2} = y_0 + \frac{2}{3}(y_1 - y_0) \tag{14}$$

$$z_{C1} = z_0 + \frac{1}{3}(z_1 - z_0) \tag{15}$$

$$z_{C2} = z_0 + \frac{2}{3}(z_1 - z_0) \tag{16}$$

After that, the axis vector for the interpolation points should be located. The rotational axis angles of the interpolation points M and N are $(B_0, A_0)$ and $(B_1, A_1)$, respectively, and their corresponding tool axis vectors are indicated by $(i_0, j_0, k_0)$ and $(i_1, j_1, k_1)$. It can be assumed that the angles for the rotational axes of interpolation points $C_{11}$ and $C_{21}$ are $(B_{c1}, A_{c1})$ and $(B_{c2}, A_{c2})$, respectively, while their corresponding tool axis vectors are represented by $(i_{c1}, j_{c1}, k_{c1})$ and $(i_{c2}, j_{c2}, k_{c2})$. The first interpolation point can transform or maintain the B axis unchanged, as well as transform the A axis or maintain the A axis unchanged. With the first interpolation point between the five-axis linkage tool center points, the B axis is kept unchanged, and the A axis transformation is taken as an example to solve the tool axis vector of the interpolation point. Then, the rotation axes corresponding to the interpolation points $C_{11}$ and $C_{21}$ are as specified in Equations (17) and (18).

$$B_{c1} = B_0,\ A_{c1} = 1/2(A_1 - A_0) + A_0 \tag{17}$$

$$B_{c2} = B_1,\ A_{c2} = 1/2(A_1 - A_0) + A_0 \tag{18}$$

After obtaining the angle of the rotation axis of the interpolation points, Equation (3) can be used to determine the tool axis vectors $(i_{c1}, j_{c1}, k_{c1})$ and $(i_{c2}, j_{c2}, k_{c2})$ that correspond to interpolation points $C_{11}$ and $C_{21}$.

### 3.3. Interpolation Detection

To confirm the viability of the interpolation program, the impeller shunted blade's machining trajectory is produced by the UG12.0 software. The interpolation algorithm is applied to realize the process section and interpolates the tool position data to verify whether the five-axis machine tool code has five-axis linkage coordinates. Partial machine tool coordinate data for the blade track processing are displayed in Table 2, while Table 3 displays the corresponding machine tool coordinate data after transformation. The conversion from five-axis linkage coordinates to four-axis linkage machine coordinates is achieved via interpolation, as can be observed from the data in the tables.

**Table 2.** Machine tool coordinates before interpolation.

| X | Y | Z | B | A |
|---|---|---|---|---|
| 150.891 | −1.827 | −38.483 | 40.823 | 336.463 |
| 149.828 | −1.885 | −37.6 | 40.5 | 336.789 |
| 148.746 | −1.942 | −36.712 | 40.173 | 337.115 |
| 147.649 | −1.998 | −35.824 | 39.844 | 337.438 |
| 146.552 | −2.048 | −34.946 | 39.516 | 337.756 |

**Table 3.** Machine tool coordinates after interpolation.

| X | Y | Z | B | A |
|---|---|---|---|---|
| 150.891 | −1.827 | −38.483 | 40.823 | 336.463 |
| 150.915 | −1.871 | −38.515 | 40.823 | 336.626 |
| 149.805 | −1.841 | −37.568 | 40.5 | 336.626 |
| 149.828 | −1.885 | −37.6 | 40.5 | 336.789 |
| 149.852 | −1.929 | −37.633 | 40.5 | 336.952 |
| 148.722 | −1.899 | −36.68 | 40.173 | 336.952 |
| 148.746 | −1.942 | −36.712 | 40.173 | 337.115 |
| 148.769 | −1.985 | −36.745 | 40.173 | 337.276 |
| 147.626 | −1.955 | −35.792 | 39.844 | 337.276 |
| 147.649 | −1.998 | −35.824 | 39.844 | 337.438 |
| 147.672 | −2.038 | −35.856 | 39.844 | 337.597 |
| 146.529 | −2.007 | −34.914 | 39.516 | 337.597 |
| 146.552 | −2.048 | −34.946 | 39.516 | 337.756 |

The nonlinear error is a unique type of error associated with five-axis CNC machine tools and is a significant factor contributing to the machine's geometric errors. Both five-axis and four-axis linkage setups exhibit nonlinear errors due to the rotational shaft's influence. The nonlinear errors of the trajectories are carefully analyzed and compared both before and after interpolation to determine their impact on the machining process.

Figure 8 delineates the interpolation intention tool of the five-axis CNC machine tool, illustrating the intention behind the interpolation process. It is noteworthy that nonlinear error typically manifests as a three-dimensional error that can be challenging to express and resolve. However, in this study, the nonlinear error is represented as a two-dimensional graph, making it more convenient to understand and analyze. The tool position data of the adjacent tool position points M and N are respectively $(x_0, y_0, z_0, i_0, j_0, k_0)$ and $(x_1, y_1, z_1, i_1, j_1, k_1)$, and its corresponding machine tool coordinates are $(X_0, Y_0, Z_0, B_0, A_0)$ and $(X_1, Y_1, Z_1, B_1, A_1)$. L(t) represents the ideal interpolation trajectory, while Q(t) represents the actual track of the machine tool, which is subject to the effects of the rotation axis.

Therefore, Equation (19) establishes the equation of the interpolation trajectory using time t as a parameter, accounting for the influence of the rotational axis on tool motion.

$$
\begin{cases}
X(t) = X_0 + (X_1 - X_0) \times t \\
Y(t) = Y_0 + (Y_1 - Y_0) \times t \\
Z(t) = Z_0 + (Z_1 - Z_0) \times t \quad (0 \leq t \leq 1) \\
B(t) = B_0 + (B_1 - B_0) \times t \\
A(t) = A_0 + (A_1 - A_0) \times t
\end{cases}
\tag{19}
$$

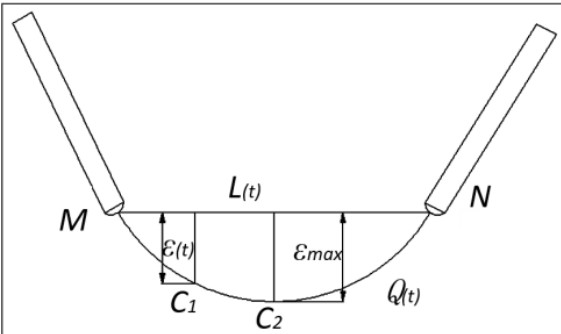

**Figure 8.** Description of the nonlinear error.

The maximum deviation amount $\varepsilon_{max}$ of Q(t) from the ideal tool point trajectory L(t) between adjacent tool points can be estimated as the nonlinear error. The actual tool tip trajectory Q(t) is determined using the machine tool's kinematic transformation model, following the process outlined below. Firstly, the direction vector of the interpolated straight line L(t) is $\vec{a}$. Subsequently, the distance between any point on Q(t) and L(t) can be determined by Equation (20).

$$
\varepsilon(t) = \frac{\left| \vec{a} \times \vec{AC_1} \right|}{\left| \vec{a} \right|}
\tag{20}
$$

The location of the maximum nonlinear error usually occurs near the midpoint of two adjacent tool path interpolation segments. After considering the impact of tool radius and corner changes on nonlinear errors, point D was established as the contact point of the tool. The connection between tool contacts is parallel to plane *XOY*, with point O being the center point of the tool. Additionally, angle *A* moves from angle $A_0$ to $A_1$.

As a result of the orthogonal design of the five-axis CNC machine, the movement of its two rotating axes can be considered relatively independent. By calculating the nonlinear errors produced by each rotating shaft in their respective moving planes, the nonlinear error of the whole five-axis NC machine is obtained. The nonlinear error caused by the motion change of the *A* axis on the plane *YOZ* is analyzed here as an example.

The $M_0$ coordinates of the tool center point of the tool site M are $(Y_m, Z_m)$, and the $N_0$ coordinates of the tool center point of the tool site N are $(Y_n, Z_n)$. The coordinates of the tool contact D1 are $(Y_{d1}, Z_{d1})$, and the coordinates of the tool contact D2 are $(Y_{d2}, Z_{d2})$. The linear equation of the line between M and N of the center of the tool is as follows:

$$
Z_0 = (Y_0 - Y_m) \frac{Z_n - Z_m}{Y_n - Y_m} + Z_m
\tag{21}
$$

By examining the relationship between the position of the tool contact and the point of the tool center, the linear equation of the contact line be obtained. Given that many current



CNC systems utilize linear interpolation, the rotation of the *A* angle also follows a linear progression, and the equation regarding the angle of the A axis can be obtained.

$$A = (Y_0 - Y_m)\frac{A_1 - A_0}{Y_n - Y_m} + A_0 \tag{22}$$

The Z component of the tool contact is obtained using Equations (21) and (22).

$$Z = Z_0 - r \times \sin(A) \tag{23}$$

Equation (23) takes the tool radius *r* into account. Consequently, the following equation is derived for the deviation of Z.

$$\frac{d_z^2}{d_y^2} = r\left(\frac{A_1 - A_0}{Y_n - Y_m}\right)^2 \sin A \tag{24}$$

When Equation (24) is equal to zero, It can be assumed that changes in the program's angle are typically small. As a result, we can derive Equations (25) and (26).

$$\cos A \approx \cos\left(\frac{A_0 + A_1}{2}\right) \tag{25}$$

$$\varepsilon_{Amax} \approx \frac{r(A_1 - A_0)^2 \left|\sin\frac{(A_1 + A_0)}{2}\right|}{8} \leq \frac{r(A_1 - A_0)^2}{8} = r(\Delta A)^2/8 \tag{26}$$

The maximum nonlinear error caused by the B axis in the same way is $\varepsilon_{bmax} = r(\Delta B)^2/8$. As the A and B axes are orthogonal and the maximum nonlinear errors typically appears at the midpoint of program segments, the maximum nonlinear error of the machine tool will occur at the midpoint of the program segment. Furthermore, the maximum nonlinear error equation of machine tool can be approximately obtained, using Equation (27). The r is tool radius. The $\Delta A$ is the angle variables of the A axis of adjacent points. The $\Delta B$ is the angle variables of the B axis of adjacent points.

$$\varepsilon_{max} \approx \sqrt{\varepsilon_{amax}^2 + \varepsilon_{bmax}^2} = r\sqrt{\left(\Delta A^4 + \Delta B^4\right)}/8 \tag{27}$$

The maximum nonlinear error $\varepsilon_{1max}$ between the five-axis points M and N before interpolation is $r\sqrt{\left(\Delta A^4 + \Delta B^4\right)}/8$. The maximum nonlinear error $\varepsilon_{2max}$ between the tool path point M and the interpolation point $C_{11}$ after interpolation is $r\sqrt{\left(\frac{1}{2}\Delta A\right)^4}/8$. The maximum nonlinear error $\varepsilon_{3max}$ between the interpolation points $C_{11}$ and $C_{21}$ after interpolation is $r\sqrt{\left(\Delta B^4\right)}/8$. The maximum nonlinear error $\varepsilon_{4max}$ between the interpolation point $C_{21}$ and the tool path point N after interpolation is $r\sqrt{\left(\frac{1}{2}\Delta A\right)^4}/8$. Then, $\varepsilon_{1max} > \varepsilon_{2max}$, $\varepsilon_{1max} > \varepsilon_{3max}$, $\varepsilon_{1max} > \varepsilon_{4max}$.

The nonlinear error is a critical cause of geometric error in five-axis CNC machines. Reducing nonlinear error is an effective approach to improve the machining accuracy of the parts.

## 4. Experimental Verification

### 4.1. Experimental Preparation

To verify the viability of the four-axis linkage machining strategy for the five-axis CNC machine, the actual machining experiment of the impeller parts is carried out. The impeller

used in this experiment has an outer diameter of 72 mm and comprises five sets of blades and splitter blades. The 3D model is shown in Figure 9.

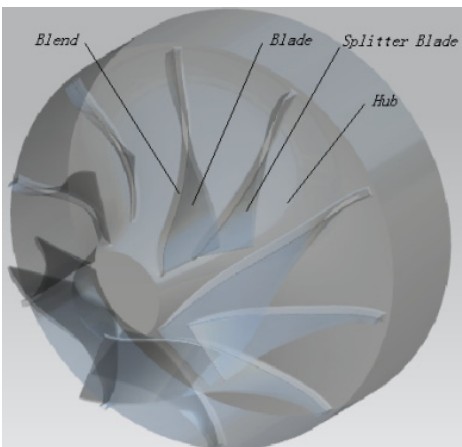

**Figure 9.** Impeller model.

The machine tool utilized in this study is a five-axis vertical machining center with a swinging head and a rotary table. Based on the structural features of the machine tool, the processing flow of the impeller is worked out. Table 4 provides details of the machining process as well as the cutting parameters employed during the impeller fabrication.

**Table 4.** Process flow and cutting parameters of impeller machining.

| Process Name | Tool | Speed (r/min) | Feed (mm/pm) |
|---|---|---|---|
| Turning | Outer circle tool | 500 | 50 |
| Multi-blade rough | ⌀4 ball-end tool | 7000 | 250 |
| Blade finishing | ⌀3 ball-end tool | 8000 | 400 |
| Splitter blade finishing | ⌀3 ball-end tool | 8000 | 400 |
| Wheel hub finishing | ⌀3 ball-end tool | 8000 | 400 |
| Blade and splitter blade blend | ⌀1 ball-end tool | 7000 | 250 |

Firstly, the blank was made by utilizing aluminum parts. The basic shape of the rotating body was created through NC turning in order to ensure that the meridian accuracy was maintained.

Secondly, the tool selected for rough machining through to finish machining was dependent on the distance between the blades and the chamfering requirements. A ball head tool or taper ball head tool with different specifications can be adopted. For the impeller being machined in this study, two blade impeller machining ball tools of R1.5 and R2 were adopted for impeller fabrication. Thirdly, the impeller model was imported into UG. According to Table 4, the appropriate processing procedure for the impeller was selected, and reasonable parameters were selected to generate the path for the five-axis tool. Thirdly, the impeller model was introduced into the UG. According to Table 4, the processing procedure of the impeller was selected, and reasonable parameters were selected to generate the five-axis tool path.

According to the interpolation algorithm, the tool position data for each trajectory were interpolated in MATLAB, resulting in the creation of a new four-axis linkage tool position file. This file was subsequently imported into UG. The converted tool trajectory can be observed in Figure 10.

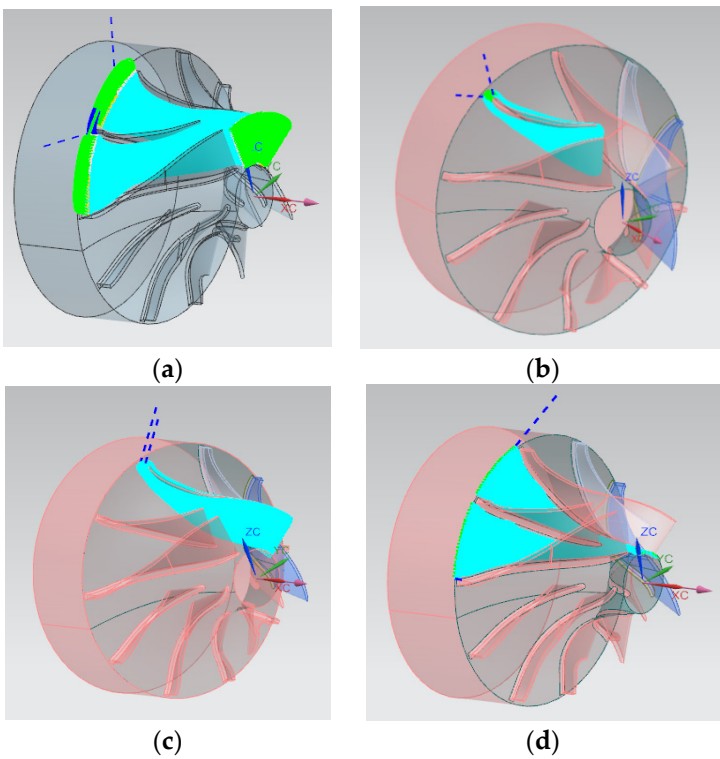

**Figure 10.** Impeller machining trajectory. (**a**) Multi-blade rough; (**b**) splitter blade finishing; (**c**) blade finishing; (**d**) wheel hub finishing.

### 4.2. Results Analysis

Firstly, the four-axis linkage machining trajectory of the impeller was simulated using the UG12 software's simulation module for machine tool processing. The purpose of this simulation was to check for any tool interference issues. After the simulation, there were no issues of cutting tool interference or other problems in the four-axis machining trajectory, and practical processing tests could be carried out.

Secondly, to analyze and confirm the accuracy of the impeller blade machining, the adjacent five-axis five-linkage points were randomly selected for testing purposes. Table 5 displays the tool position data for the adjacent tool center points prior to interpolation, while Table 6 shows the interpolated tool position data.

**Table 5.** Tool position data before interpolation.

| X | Y | Z | I | J | K |
|---|---|---|---|---|---|
| −18.0272 | −9.2689 | 18.6089 | 0.3593827 | −0.2294031 | 0.9045542 |
| −18.1461 | −9.3634 | 18.6306 | 0.3655032 | −0.2311342 | 0.9016565 |

**Table 6.** Tool position data after interpolation.

| X | Y | Z | I | J | K |
|---|---|---|---|---|---|
| −18.0272 | −9.2689 | 18.6089 | 0.3593827 | −0.2294031 | 0.9045542 |
| −18.066833 | −9.300400 | 18.616133 | 0.365500 | −0.229973 | 0.901955 |
| −18.106467 | −9.331900 | 18.623367 | 0.359382 | −0.230561 | 0.904260 |
| −18.1461 | −9.3634 | 18.6306 | 0.3655032 | −0.2311342 | 0.9016565 |

Analyzing the data in Tables 5 and 6, it can be calculated that the maximum non-linear errors before and after transformation of the five-axis five-linkage trajectory were $\varepsilon_{max1} = 1.4$ μm and $\varepsilon_{max2} = 0.36$ μm, respectively. It can be deduced that the maximum

nonlinear error of the adjacent points after interpolation is less than the maximum nonlinear error prior to interpolation, and the result is $\varepsilon_{max1} > \varepsilon_{max2}$. Therefore, this algorithm demonstrated its ability to effectively enhance the machining quality of parts. However, it is essential to further improve the method. Since the addition of tool position occurs after the interpolation of the adjacent five-axis tool position, the processing time is increased, and processing efficiency is sacrificed as a result.

The results of 3D IPW were compared between the five-axis path (finishing path) calculated by the impeller programming module and the path generated by the transformation algorithm. Figure 11 shows that the machining accuracy of the blade (1) and hub finishing is relatively similar, with slightly better accuracy observed on the blade. When the path transformed by the algorithm is used to process the root of the blade (3) and the splitter blade (2), there are many places where the machining accuracy deteriorates. This may be due to significant changes in the tool vector, resulting in missing or over-cutting. From the results of Figure 11b,d, it is clear that, prior to using the algorithm, the maximum machining error position in the impeller machining was located at the root of the splitter blade, with an error value of 0.201 mm. After the algorithm was applied, the maximum machining error position remained located at the root of the splitter blade, with a modified error value of 0.479 mm.

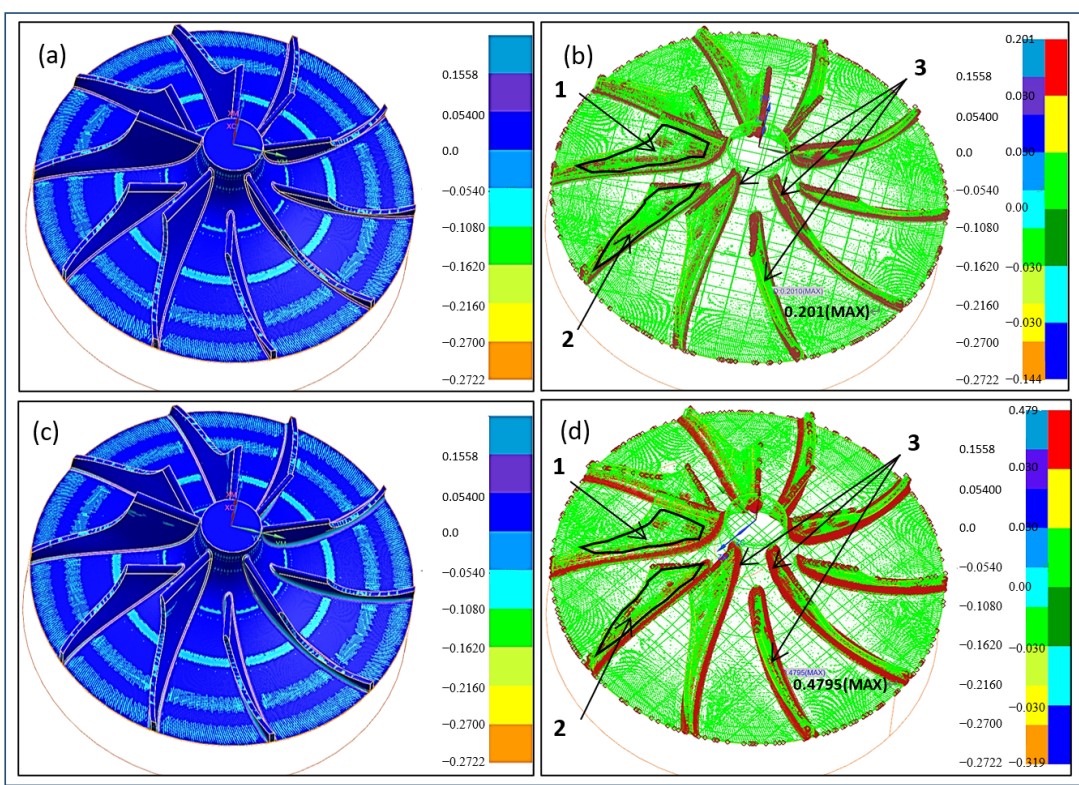

**Figure 11.** Analysis results of (**a**) 3D IPW for five-axis five-linkage, (**b**) deviation gauge for five-axis five-linkage, (**c**) 3D IPW for five-axis four-linkage, (**d**) deviation gauge for five-axis four-linkage.

Table 7 summarizes the error report obtained by the deviation gauge. Using the five-axis five-linkage trajectory, a total of 614,723 sampling points were analyzed. Among them, 47,215 points were found to be out of tolerance, accounting for 7.68% of the total sampling points. After applying the algorithm in this study and analyzing the data using the five-axis four-linkage trajectory, a total of 597,861 sampling points were obtained, out of which 68,294 points were found to be out of tolerance, accounting for 11.42% of the total sampling points. This mainly manifested at the root of the blade and splitter blade.

**Table 7.** Error reporting of deviation gauge.

| Five-Axis Five-Linkage | | | Five-Axis Four-Linkage | | |
|---|---|---|---|---|---|
| | Number | Percentage | | Number | Percentage |
| Total samples | 614,723 | 100.00 | Total samples | 597,861 | 100.00 |
| Inside inner tolerance | 567,508 | 92.32 | Inside inner tolerance | 529,567 | 88.58 |
| Inside outer tolerance | 567,508 | 92.32 | Inside outer tolerance | 529,567 | 88.58 |
| Out of tolerance | 47,215 | 7.68 | Out of tolerance | 68,294 | 11.42 |

Based on the analysis presented above, two key observations can be made regarding the proposed interpolation algorithm. Firstly, it enables the transformation from five-axis five-linkage machining to five-axis four-linkage machining. Secondly, the algorithm demonstrates superior machining accuracy in surface machining operations where the tool axis vector changes minimally. However, when the tool axis vector undergoes significant changes, the machining accuracy of the algorithm may experience a relative decline.

*4.3. Machining Verification*

The validation of the feasibility of the proposed strategy and interpolation algorithm is demonstrated through the successful machining of the impeller parts. Furthermore, the developed post-processing program for the five-axis vertical NC machining center allows for the generation of NC machining programs that are executable on machine tools. Figure 12 presents the machined impeller parts, which serve as a testament to the high-precision results attainable using the proposed approach. The significance of these results lies in their potential to reduce manufacturing costs while still maintaining superior standards of machining accuracy for complex surfaces in the manufacturing industry. In conclusion, this paper constitutes a noteworthy contribution to the advancement of efficient machining strategies for contemporary manufacturing industries.

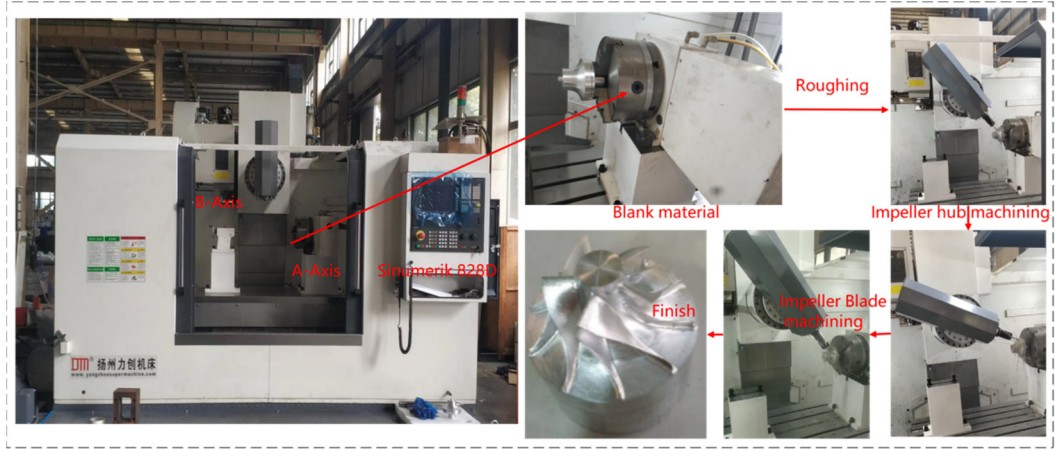

**Figure 12.** Finishing impeller.

**5. Conclusions**

The results of this paper demonstrate the effectiveness of the proposed strategy and interpolation algorithm for four-axis machining using a five-axis machine tool. The kinematics modeling and analysis provide a basis for understanding the relationship between tool position data and machine axis movements, which is essential for developing effective machining strategies. The four-axis machining strategy proposed in this paper offers a cost-effective alternative to five-axis machining while still providing high-precision results. It can be seen from the IPW analysis results that the machining accuracy of the five-axis and four-linkage trajectory after transformation is basically the same as that before transformation. Furthermore, by incorporating the deviation gauge option, it was found that the maximum machining error position of the impeller machining before and after using the

algorithm were both located at the root of the splitter blade, with error values of 0.201 mm and 0.479 mm, respectively. The points that were found to be out of tolerance accounted for 7.68% and 11.42% of the total samples points, respectively.

Based on the analysis and machining validation, the proposed interpolation algorithm effectively transforms five-axis five-linkage machining into five-axis four-linkage machining. It is noteworthy that the algorithm exhibits higher machining accuracy in surface machining operations where the tool axis vector undergoes minimal changes.

**Author Contributions:** Conceptualization, Y.W., J.D. and J.L.; methodology, Y.W., L.J. and J.D.; software, Y.W. and J.D.; validation, Y.W., J.L., J.D., M.L. and J.L.; formal analysis, Y.W.; investigation, J.D., M.L. and J.L.; resources, Y.W., M.L. and J.L.; data curation, Y.W., L.J., J.D. and M.L.; writing—original draft preparation, Y.W., L.J. and J.D.; writing—review and editing, Y.W., L.J., M.L. and J.L.; supervision, Y.W., M.L. and J.L.; project administration, Y.W.; funding acquisition, Y.W. and M.L. All authors have read and agreed to the published version of the manuscript.

**Funding:** This research received no external funding.

**Institutional Review Board Statement:** Not applicable.

**Informed Consent Statement:** Not applicable.

**Data Availability Statement:** The data are contained within the article.

**Conflicts of Interest:** Author Liang Ji was employed by the company Beijing Shiny Technology Co., Ltd. And author Jiashang Dong was employed by the company Avic Jonhon Optronic Technology Co., Ltd. The remaining authors declare that the research was conducted in the absence of any commercial or financial relationships that could be construed as a potential conflict of interest.

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
