# Peer review of "Research on Continuous Machining Strategy for Five-Axis Machine Tool: Five-Axis Linkage to Four-Axis Linkage"

_applsci, doi:10.3390/app13127038_

Round 1

Reviewer 1 Report

What is the serious shortage facing the five-axis linkage control system?

What are the technical difficulties of a five-axis linkage control system?

Identify this work's aims and critical novelty at the end of the introduction.

What does A prime and B mean in Fig. 3 and 4?

The sentences are too long, and the information can not be followed.

What is the suitable validation metric between the four-axis and five-axis machining strategy, and how can it be compared to a final product?

Reviewer 2 Report

The paper presents a method to reduce a five-axis linkage path planning problem into a four-axis one by restricting movements about one of the rotation axes. Some issues are present in this work.

The most severe issue with the work is that it is not clear why five-axes problems are considered: in the Introduction it is stated that three-axes are insufficient for complex paths. Instead of taking the logical step and considering four-axes path plans the authors go straight to five-axes, and then propose a solution to reduce their problem to four axes. This is contradictory, and the approach considering five axes and placing an artificial restriction (instead of just considering four axes from the get-go) needs to be explained.

Less severe issues include:

1. It is not clear what is meant in the first sentence of the Abstract.

2. The impeller is an example application of the algorithm proposed in the paper. It is not central to it, nor is the work restricted to impellers alone - thus the explanations in line 19--21 should be summarised or shortened.

3. The literature review gives many different related works, but it is not clear what their relevance is to the work outside of them not using the proposed method. What shortcoming(s) in existing results does the proposed method aim to address?

4. The quality of the diagrams in the paper are also poor:
- Fix the alignment of the caption for Fig. 2, and the positions of Figs. 3--4.
- The size of the text on many of the figures (especially flowchart figures like Figs. 5--6) is too small and difficult to read.
- Cite the source of figures taken from elsewhere (e.g., Fig. 2)

There are various language issues in this paper, particularly in the portions with unclear meanings in comment point 1.

Reviewer 3 Report

In this paper, a corresponding machine tool kinematic model and interpolation algorithm were established based on theory, which converted the five-axis linkage into the application of four-axis linkage on a five-axis machine tool, and its feasibility was verified through experiments. The structure of the paper is clear, however, it still needs some minor revisions:

1. Some images are not clear enough and can be replaced with higher quality ones.

2. In "4+1" machining, there is a question of whether it is necessary to lock the rotation axis. The position of the locking point of the rotation axis is related to factors such as tool path and the shape of the workpiece, which requires certain analysis and calculation. It is recommended to optimize the interpolation algorithm.

3. In chapter four, the validation of the method is performed by machining an impeller, but lacks convincing comparative data between four-axis and five-axis linkage.

4. The conclusion is suggested to rework with more detailed data and exact results.

the language can be improved

Reviewer 4 Report

The paper presents an algorithm that allows replacing five-axis simultaneous machining with four-axis simultaneous machining with a fifth-positioned axis. The authors have presented analytical equations and an algorithm that converts a five-axis path to a "four-axis" path. A significant part of the publication is occupied by these analytical considerations. Nevertheless, despite such an extensive mathematical apparatus, the authors do not fully prove the effects of the introduced interpolation algorithm. For this reason, the part of the research in which the authors check the results of the applied algorithm on the accuracy results should be greatly expanded. At the moment, the authors have only presented a machined impeller; the same is possible without any algorithm.

1.       Absolute improvement of the study of dimensional and shape accuracy, at the stage of programming in the NX system, and present the results showing accuracy maps in the assumed tolerance. For this purpose, the authors should compare the IPW after verification with a five-axis path obtained from operations from the impeller programming module (finishing paths) with the path generated by the transformation algorithm. To do this, use the "Deviation Gauge" function which allows comparing the nominal 3D model of the impeller with the IPW model (set high resolution IPW) obtained after verification in the NX system in the manufacturing module. The comparison should be made for both programmed paths. This approach will allow you to check the accuracy of the machining of the applied rotor already at the programming stage and can confirm error considerations.

2.       Conclusion is too general, there is no specification of the effects of the algorithm used

3.       Provide the real stand on which the trials were carried out.

Round 2

Reviewer 1 Report

The authors addressed my comments.

Still the sentences are too long.

Author Response

Still the sentences are too long.

Response: Thank you very much for your review comments. We have made revisions to the paper, simplifying and breaking down long sentences as much as possible to enhance readability and clarity. Please check the revised manuscript. We appreciate your guidance and attention.

Reviewer 2 Report

The authors have clarified the scope of the work better. The responses to several comments however has been unsatisfactory. In particular, all changes to the manuscript need to be highlighted.

1. Related to the previous comment 3, there is no explanation on what strengths the proposed method has over the many different methods in the literature review. The addition in a later portion is not relevant either, and does not clarify the lack of motivation for the current method in the Introduction.

2. Related to the previous comment 4, the alignment of the caption for Fig. 2 and sources for cited figures are also not provided.

Many of the previous English errors are still present in this revised manuscript (e.g., `traditional three-axis surface processing technology has been difficult to meet').

Reviewer 4 Report

The results presented were obtained by verification using a graphical display of the distribution of machining residues. Present the results for both toolpaths separately for hub and separately for blade. It is necessary to ensure that the maps are the same within the same range of changes in the overmachining residuals. For this purpose, use high accuracy IPW generation and adopt automatic display in the first step, followed by individual settings. Remember to adopt the same color levels and extreme ranges.

Better results can be obtained using the deviation gauge option where the authors could better analyze the machined areas mentioned above.

For the moment, the results presented in this way are a mistake in the art of research

Round 3

Reviewer 2 Report

My comments have been sufficiently addressed.

There are still some hanging sentences present in the work, e.g., `Through the above analysis', `As shown in Figure 2. This...'

Author Response

My comments have been sufficiently addressed.

There are still some hanging sentences present in the work, e.g., `Through the above analysis', `As shown in Figure 2. This...'

Response: Thank you for your feedback and suggestions. We are glad to know that you believe we have addressed your comments satisfactorily. We have also revised and refined the manuscript to eliminate the hanging sentences and ensure a clearer and smoother presentation. We appreciate your thorough review and are committed to improving the quality of our work.

Reviewer 4 Report

I accept in the present form article.

Author Response

I accept in the present form article.

Response: Thank you for considering our article for publication. We appreciate your time and effort in reviewing our work. We will continue to strive for excellence in our research and writing to contribute to the advancement of our field.